



# A 1 km global cropland dataset from 10000 BCE to 2100 CE

Bowen Cao[1], Le Yu[1,2,*], Xuecao Li[3], Min Chen[4,5], Xia Li[6], Peng Gong[2,7]

[1]Ministry of Education Key Laboratory for Earth System Modeling, Department of Earth System Science, Tsinghua University, Beijing 100084, China

[2]Ministry of Education Ecological Field Station for East Asian Migratory Birds, Beijing 100084, China
[3]College of Land Science and Technology, China Agricultural University, Beijing 100083, China
[4]Department of Forest and Wildlife Ecology, University of Wisconsin-Madison, 1630 Linden Drive, Madison, WI 53706-1598, USA
[5]Nelson Institute Center for Climatic Research, University of Wisconsin-Madison, 1225 W. Dayton St. Madison, WI 53706

USA
[6]Ministry of Education Key Laboratory of Geographic Information Science, School of Geographic Sciences, East China Normal University, Shanghai 200241, China
[7]Department of Geography and Department of Earth Sciences, University of Hong Kong, Hong Kong

*Correspondence to*: Le Yu (leyu@tsinghua.edu.cn)

**Abstract.** Cropland greatly impacts food security, energy supply, biodiversity, biogeochemical cycling, and climate change. Accurately and systematically understanding the effects of agricultural activities requires cropland spatial information with high resolution and a long time span. In this study, the first 1 km resolution global cropland proportion dataset for 10000 BCE-2100 CE was produced. With the cropland map initialized in 2010 CE, we first harmonized the cropland demands

extracted from the History Database of the Global Environment 3.2 (HYDE 3.2) and the Land-Use Harmonization 2 (LUH2) datasets, and then spatially allocated the demands based on the combination of cropland suitability, kernel density, and other constraints. According to our maps, cropland originated from several independent centers and gradually spread to other regions, influenced by some important historical events. The spatial patterns of future cropland change differ in various scenarios due to the different socioeconomic pathways and mitigation levels. The global cropland area generally shows an

increasing trend over the past years, from 0 million $km^2$ in 10000 BCE grows to 2.8 million $km^2$ in 1500 CE, 6.2 million $km^2$ in 1850 CE, and 16.4 million $km^2$ in 2010 CE. It then follows diverse trajectories under future scenarios, with the growth rate ranging from 18.6% to 82.4% between 2010 CE and 2100 CE. There are large area disparities among different geographical regions. The mapping result coincides well with widely-used datasets at present in both distribution pattern and total amount. With improved spatial resolution, our maps can better capture the cropland distribution details and spatial

heterogeneity. The spatiotemporally continuous and conceptually consistent global cropland dataset serves as a more comprehensive alternative for long-term earth system simulations and other precise analyses. The flexible and efficient harmonization and downscaling framework can be applied to specific regions or extended to other land use/cover types through the adjustable parameters and open model structure. The 1 km global cropland maps are available at https://doi.org/10.5281/zenodo.5105689 (Cao et al., 2021a).



## 1 Introduction

Land use changes driven by humans have profound impacts on climate change, biogeochemical cycling, biodiversity, energy supply, and food security (Foley et al., 2005; Kalnay and Cai, 2003; Ito and Hajima, 2020; Poschlod et al., 2005). As one of the predominant land use types, agricultural land serves as the important carbon budget component and the basic elements of food production, contributing substantially to global change in both the natural environment system and the social-economic system (Friedlingstein et al., 2020; Godfray et al., 2010). In recent decades, significant progress has been made in agricultural monitoring, including cropland extents (Yu et al., 2013; Lu et al., 2020), cropland types (Cao et al., 2021b), crops (Zhong et al., 2014; Bargiel, 2017), and farming practices (Biradar and Xiao, 2011; Estel et al., 2015), providing basic and direct information to support specific research and management for specific years or periods. In comparison, simulating or analyzing the effect of cropland change from the beginning of farming to the end of this century can provide a comprehensive view for understanding agriculture, which is of great significance for establishing long-term environmental or economic strategies (Olofsson and Hickler, 2008; Pongratz et al., 2009; Molotoks et al., 2018; Zabel et al., 2019). However, big gaps and uncertainties remain in quantifying the long-term global change through models and other geospatial analysis methods, largely affected by the input land use/cover data (Prestele et al., 2017). Therefore, accurate global cropland change information, especially a harmonized cropland dataset at high resolution from past to future, plays a crucial role in improving the simulation accuracy and supporting the detailed analysis.

Some efforts have been made in developing historical or future cropland products till now. In reconstructing the spatial distribution of past cropland, the representative products at global-scale include the Sustainability and the Global Environment (SAGE) dataset (5'×5' resolution for global cropland during 1700 CE-1992 CE) (Ramankutty and Foley, 1999), the Millennium Land Cover Reconstruction (ML08) dataset (0.5°×0.5° resolution for global agricultural areas during 800 CE-1992 CE) (Pongratz et al., 2008), the Kaplan and Krumhardt (KK10) dataset (5'×5' resolution for 8000 BCE-1850 CE) (Kaplan et al., 2011), and the History Database of the Global Environment (HYDE) dataset (Klein Goldewijk, 2001; Klein Goldewijk et al., 2010, 2017). Among these datasets, the HYDE dataset is constantly updated, and the latest version (HYDE 3.2) achieves the highest spatial resolution (5'×5' resolution) and the longest time-span (10000 BCE-2017 CE) through a more comprehensive and reasonable algorithm (Klein Goldewijk et al., 2017). The above global-scale datasets usually employed a "top-to-bottom" method to downscale the historical records of cropland area according to the cropland suitability, population, and current distribution of cropland. Additionally, there are some regional or local products with higher resolution (Fuchs et al., 2013; Yang et al., 2015; Yu and Lu, 2018).

Research on simulating future land use/cover change is booming in recent years, with cropland as one of the types. To prepare for each generation of the World Climate Research Program Coupled Model Intercomparison Project (CMIP), Integrated Assessment Model (IAM) teams constructed a series of scenarios for future land use projections as inputs of Earth System Models (ESMs) (O'Neill et al., 2016; Riahi et al., 2017). Based on these scenarios, the Land-Use Harmonization (LUH) project provided the harmonized land use datasets as a part of CMIP, including LUH (0.5°×0.5° resolution for 1500



CE-2100 CE) (Hurtt et al., 2011) corresponding to CMIP5 and LUH2 (0.25°×0.25° for 850 CE-2100 CE) (Hurtt et al., 2020) corresponding to CMIP6. More and more studies are devoted to future land use simulations at a higher resolution based on various scenarios to satisfy finer applications. On the global scale, there are several 1 km resolution datasets (Li et al., 2016; Li et al., 2017; Cao et al., 2019). The resolutions of some local or regional downscaling researches were even finer (Chen et al., 2018; Wang et al., 2018; Xu et al., 2015). Spatially explicit land use/cover change models such as Cellular Automata (CA) (White et al., 1997), Future Land Use Simulation (FLUS) (Liu et al., 2017), Conversion of Land Use and its Effects (CLUE) (Veldkamp and Fresco, 1996), Agent-based model (ABM) (Matthews et al., 2007), and Demeter (Vernon et al., 2018), were extensively adopted in the future land use simulation studies.

Although the above datasets are widely used and contribute a lot to the related research, the existing global products are relatively scarce and have large uncertainty (Klein Goldewijk and Verburg, 2013; Prestele et al., 2016; Alexander et al., 2017). Huge differences between these datasets make them difficult to be well connected and even cause contradictions in applications (Prestele et al., 2017). With the development of related models and analytical methods, there is a growing demand for continuous datasets from past to future. So far, only the LUH project provided the spatiotemporally continuous global land use datasets throughout history and future. Nevertheless, although the resolution has been improved to 0.25° in the latest version, LUH2 is still too coarse to describe the details of cropland distribution and support the accurate analysis (Schaldach et al., 2011; Liao et al., 2020). Underestimation and overestimation are inevitably caused when using low resolution land use/cover datasets (Yu et al., 2014), which decrease the credibility of the related research results greatly. The low-resolution problem is also common in many other global-scale datasets mentioned above. Besides, since agriculture approximately originates in 10000 BCE, the important initial period of cropland development is omitted in LUH2. Therefore, a harmonized dataset from past to future with higher resolution and longer time-span is urgently needed.

As for the methods, in the above studies, historical reconstruction and future projection usually adopted different models, which cannot be merged directly. Although they performed well in producing the datasets for specific periods and regions, most of them have limited extendibility in time scale or spatial scale. Some downscaling models are simple and cannot accurately characterize the long-term change, while some other models need high computational cost to implement at a large scale and fine resolution (Council, 2014). A model capable of reflecting the cropland change in time-space consistency and high efficiency is thus required. A flexible and efficient framework for harmonizing and downscaling the cropland distribution from past to future at large-scale and high-resolution should be established.

In the study, we produced a global cropland percentage map at 1 km resolution from 10000 BCE to 2100 CE based on our proposed harmonization and downscaling framework. And we identified the cropland distribution patterns, estimated the cropland areas, and compared the mapping results with other datasets.



## 2 Method

The framework of producing the 1 km global cropland dataset for 10000 BCE-2100 CE included demand harmonization and
100  spatial downscaling (Fig.1). Details for the mapping procedure are provided in the following sections.

### 2.1 Cropland demands harmonization

Cropland demands for history and future were estimated based on the existing products. Historical demands during 10000
BCE-2010 CE were based on HYDE 3.2, which provides a complete historical land use reconstruction and serves as a basis
of long-term global change analysis and simulation (Klein Goldewijk et al., 2017). Considering that the cropland area in
105  HYDE 3.2 referred to the national statistics from the Food and Agriculture Organization (FAO) and subnational statistics of
some larger countries (Klein Goldewijk et al., 2017), the downscaling regions in this study were divided using the provincial
boundaries for several largest countries (countries with an area of >2.5 million $km^2$, i.e., Russia, Canada, China, America,
Brazil, Australia, India, Argentina, Kazakhstan) and national boundaries for the other countries. The future demands during
2010 CE-2100 CE came from the total areas of all five crop types (including C3 annual crops, C3 perennial crops, C4 annual
crops, C4 perennial crops, and C3 nitrogen-fixing crops) in the LUH2 dataset, which was consistent with the design of
CMIP6 and widely used in ESM simulations (Hurtt et al., 2020). All the eight scenarios with combinations of five Shared
Socioeconomic Pathways (SSPs) and seven Representative Concentration Pathways (RCPs) (SSP1-RCP1.9, SSP1-RCP2.6,
SSP2-RCP4.5, SSP3-RCP7.0, SSP4-RCP3.4, SSP4-RCP6.0, SSP5-RCP3.4OS, SSP5-RCP8.5) were projected in the study.

To drive the downscaling, an initial cropland map with the target resolution is indispensable. Here we selected the global
synergy cropland map for 2010 CE produced by Lu et al. (2020) as the start point. The map was generated based on the
fusion of multiple existing cropland maps and multilevel statistics of cropland area through the Self-adapting Statistics
Allocation Model (SASAM). It had higher accuracy than other mainstream datasets and better consistency with the cropland
statistics (Lu et al., 2020). And it used FAO's definition of cropland, i.e., "arable lands and permanent crops" (FAO, 2021),
which was thus inherited into our study. In preprocessing, we first aggregated map resolution from the original 500 m to the
target 1 km. Then, considering that except cropland, urban and water/snow/ice would be involved in our downscaling rules,
we supplemented maps of these two land cover types from the other dataset. One of the input data for producing the global
synergy cropland map, Globeland30 2010 (Chen et al., 2014), was selected here for its high accuracy and consistent year. To
be compatible with urban and water/snow/ice distribution extracted from Globeland30 2010, we further processed the
cropland map to ensure enough spare space is left for these types in each pixel. The non-cropland percentage in the global
synergy cropland map was increased to the sum of the urban and water/snow/ice percentage in each 1 km pixel where the
former was less than the latter. The preprocessed initial cropland map was taken as the base map for the following
procedures.

Due to the difference in methods and class definitions, there are obvious discrepancies in the cropland areas of HYDE 3.2,
LUH2, and the base map in some regions. To avoid further errors caused by the inconsistencies, harmonizing the amount is
thus necessary. We first adjusted the cropland demands in base year (2010 CE) to keep it consistent with the cropland area of





the base map. We then calculated the harmonized demands for different regions in historical and future years according to the original cropland area change rates of HYDE 3.2 and LUH2. The harmonization process can be expressed as:

$$A_{H,r,2010CE} = A_{Base,r,2010CE} \text{,} \tag{1}$$

$$A_{H,r,t} = A_{H,r,t-1} \times \frac{A_{HYDE\,3.2,r,t}}{A_{HYDE\,3.2,r,t-1}} \text{ (t=2000 CE, 1990 CE, ..., 9000 BCE, 10000 BCE),} \tag{2}$$

$$A_{H,r,t} = A_{H,r,t-1} \times \frac{A_{LUH2,r,t}}{A_{LUH2,r,t-1}} \text{ (t=2020 CE, 2030 CE, ..., 2090 BCE, 2100 CE),} \tag{3}$$

where $A_{H,r,t}$ is the harmonized cropland area for region $r$ at time step $t$ (the time step intervals are 1000 years for 10000 BCE-1 CE, 100 years for 1 CE-1700 CE, and 10 years for 1700 CE-2100 CE), $A_{Base,r,2010CE}$ is the cropland area of the base map for region $r$ in 2010 CE, $A_{HYDE\,3.2,r,t}$ and $A_{LUH2,r,t}$ are the cropland area of HYDE 3.2 and LUH2 for region $r$ at time step $t$, respectively. Namely, the demands began with the base map but followed the change trajectories of HYDE 3.2 during

10000 BCE-2010 CE and LUH2 during 2010 CE-2100 CE. The new set of cropland demands after harmonization were prepared as inputs for the spatial downscaling.

**2.2 Cropland demands downscaling**

The spatial downscaling was performed based on the developed framework as below. First, the maximum area for cropland allocation in each 1 km×1 km grid cell was determined by the following rules. For the whole downscaling period (10000

BCE-2100 CE), water and snow/ice areas were assumed constant over time and thus could not be occupied by cropland. For the future period (2010 CE-2100 CE), cropland would not expand to the urban area due to urbanization usually being regarded as the most irreversible anthropic activities (Grimm et al., 2008; Wu, 2014), and cropland located within the protected area (defined by World Database on Protected Areas (WDPA) (UNEP-WCMC. and IUCN., 2021)) could only unidirectionally change (i.e., reduce) since the base year.

Cropland is more likely to occur in the places where both natural environment and socioeconomic conditions are suitable. The spatial heterogeneity of conditions can be indicated by the suitability layers. In the study, we generated the suitability layers using the random forest (RF) regression model, which proved to be effective and efficient in dealing with the statistical problems (Breiman, 2001). Given the availability of related variables for model training and prediction, we selected eight biophysical and one socioeconomic variables to describe the key information affecting cropland suitability,

including terrain, climate, soil, and population (Table 1). Note that these variables may not be comprehensive, but they are representative and reflect different heterogeneous aspects related to cropland distribution. In large-scale research, biome or ecoregion frameworks were frequently adopted to better characterize various vegetation-climate association patterns in different regions (Yu et al., 2016). In the study, we used the World Wildlife Fund (WWF) biome system (Olson et al., 2001) to divide the world into 14 biome regions for separate model training and prediction. About half a million (0.52 million,

accounting for 0.3% global land pixels at 1 km resolution) training samples were randomly collected worldwide from the





global synergy cropland map which had been aggregated to 1 km resolution. The key parameter of the RF model, i.e., tree number, was set to 100. In the study, two types of suitability layers were generated for later use: one only relied on the biophysical drivers and another relied on both biophysical and socioeconomic drivers.

Over the long downscaling time period, cropland suitability rules change due to the various interaction patterns between
natural environment, social-economic factors, and agricultural activities. Referring to HYDE 3.2 (Klein Goldewijk et al., 2017), we divided the whole downscaling process into several periods. According to the characteristics of different periods, we made some combinations and adjustments based on the above two RF-based suitability layers to get the final cropland suitability. During the early stage of agricultural development (10000 BCE-1500 CE), limited by traditional farming practices and weak global links, cropland distribution was highly related to local population distribution. Local cropland area
was almost proportional to the population size, meanwhile, influenced by the biophysical suitability. In the last ~500 years (1500 CE-2010 CE), with the development of society, the relationship between population and cropland distribution became more and more similar with the contemporary pattern, which can be accurately characterized by machine learning model. The above RF-based biophysical-socioeconomic suitability layer was thus used for this period. In recent years, population demand in agricultural activities is weakening under population intensification and technology development (Goklany, 2009).
The impact of population on cropland distribution is negligible for most regions in the future years (2010 CE-2100 CE). Therefore, the future cropland suitability is only indicated by the biophysical suitability layer in the study. We set the dynamic time-dependent weights to connect the cropland suitability in different time periods. The final cropland suitability is formulated as follows:

$$S_{gc,t} = \sqrt{W1_{gc,t} \times S1_{gc,t} \times POP_{gc,t} + W2_{gc,t} \times S2_{gc,t} + W3_{gc,t} \times S1_{gc,t}} \, , \qquad (4)$$

where $S_{gc,t}$ represents the final cropland suitability of grid cell $gc$ at time step $t$, and $POP_{gc,t}$ represents the normalized population. $S1_{gc,t}$ and $S2_{gc,t}$ are the biophysical suitability and biophysical-socioeconomic suitability layers based on the RF model, respectively. $W1_{gc,t}$, $W2_{gc,t}$ and $W3_{gc,t}$ are time weights. Referring to the weight setting in HYDE 3.2 (Klein Goldewijk et al., 2017), $W1_{gc,t}$ is set to zero in 2000 CE and 100% in 1500 CE (and the pre-1500 CE period as well), while $W2_{gc,t}$ is set to zero in 1500 CE (and the pre-1500 CE period as well) and 100% in 2000 CE, and both $W1_{gc,t}$ and $W2_{gc,t}$
change linearly during 1500 CE-2010 CE. In the future time (after-2010 CE), $W3_{gc,t}$ is set to 100%, while $W1_{gc,t}$ and $W2_{gc,t}$ are set to zero.

Our downscaling algorithm also considered the cell states and neighborhood effects during the allocation process. It was implemented under the assumption that cropland would appear close to where it already was at the next time step. Kernel density was computed to represent the density and proximity of cropland within or around a given grid cell. It is defined by:

$$KD_{gc,t} = \frac{\sum_1^{D^2} CP_{gc,t-1}}{D^2} \, , \qquad (5)$$





where $KD_{gc,t}$ is the kernel density of grid cell $gc$ at time step $t$, and $CP_{gc,t-1}$ is the cropland proportion of the grid cell at the last time step $t-1$. A user-defined parameter, $D$, stands for the moving window size used to compute kernel density.

The above final cropland suitability and kernel density were combined to the final probability-of-occurrence layer:

$$P_{gc,t} = \sqrt{S_{gc,t} \times KD_{gc,t}}, \tag{6}$$

where $P_{gc,t}$ denotes occurrence probability of cropland in grid cell $gc$ at time step $t$. The cropland demands were then tentatively distributed to each grid cell according to the occurrence probability:

$$TA_{gc,t} = A_{H,r,t} \times \frac{P_{gc,t} \times GA_{gc}}{\sum_1^n (P_{gc,t} \times GA_{gc})}, \tag{7}$$

where $TA_{gc,t}$ is the tentative allocation of cropland area on grid cell $gc$ at time step $t$, $GA_{gc}$ is grid cell area, $A_{H,r,t}$ is the harmonized cropland demand of the region where grid cell $gc$ locates, and $n$ is the total number of grid cells in the region.

The actual allocation area on each grid cell is the minimum value of the tentative allocation area and the maximum area for cropland allocation:

$$CA_{gc,t} = min\ (TA_{gc,t} | MA_{gc,t}), \tag{8}$$

$$CP_{gc,t} = \frac{CA_{gc,t}}{GA_{gc}}, \tag{9}$$

where $CA_{gc,t}$ and $CP_{gc,t}$ are the actual allocation area and proportion on grid cell $gc$ at time step $t$. $MA_{gc,t}$ is the maximum

area for cropland allocation of the grid cell.

On the whole, cropland was continually shrinking from present to past, and more cropland tends to intensify in recent years (Hu et al., 2020). As a research at the global scale, we thus assumed that as much cropland as possible first appeared in the grid cells with cropland in the last time step. In the study, we referred to the previous research (Le Page et al., 2016) and divided the whole allocation process into two stages. In the first stage, demands were allocated to the grid cells where

cropland already existed, referred to as intensification. In the second stage, the unallocated demands were allocated to other grid cells, referred to as expansion. We used the parameter representing the moving window size for kernel density calculation, $D$, to directly control the two stages. It was set to 1 and 3 to achieve intensification and expansion in the two stages, respectively. In both stages, the allocation was processed iteratively until there was no more spare space for cropland allocation or the unallocated demands were less than the threshold (0.01 km$^2$) in all regions.

**3 Results**

**3.1 Downscaled cropland maps**

Based on the above mapping framework, we produced the 1 km resolution global cropland proportion dataset for 10000 BCE-2100 CE. Fig. 2 shows the downscaled cropland maps in several key historical years, providing an overall perspective





on the cropland change. Agriculture originated in several independent regions, including the Yangtze River Basin and
Yellow River Basin in China, Neil Delta in Africa, Middle East, some areas in Central and South America, and some
Mediterranean coastal countries (Fig. 2a). Under the restriction of natural and socioeconomic conditions, it slowly spread to
other places at different speeds (Fig. 2b and 2c). Until 1500 CE, agriculture was prevalent throughout China, India, western
Europe, Middle East, Central America, and Africa (Fig. 2d). Since the Great Geographical Discoveries strengthened the
global trade links, agricultural development was thus strengthened (Fig. 2e). However, due to the dramatic population
declines and political devastation caused by colonialism and disease during the Age of Exploration, cropland was still scarce
in North and South America and Oceania. In the 19th century, many countries speeded up the social development rate, and
western countries carried out the large-scale Industrial Revolution and introduced machinery into agricultural production,
which made agricultural development leap forward and caused a great acceleration in cropland area and production. The area
increase was obvious worldwide (Fig. 2f and g). Moreover, because of the independence of North American countries, the
cropland area there rose rapidly during the 100 years (Fig. 2f and g). In the last century, owing to great development after the
successive independence of South America, Australia, and other colonies, the cropland in the southern hemisphere expanded
evidently. With the continuous social development, the cropland in the northern hemisphere also increased on the original
basis (Fig. 2g and h).

The downscaled cropland maps in 2100 CE under the eight scenarios are displayed in Fig. 3. Since the scenarios vary in
terms of socioeconomic pathways and mitigation levels, the global cropland distribution patterns are widely diverse. In most
regions worldwide, cropland has relatively small changes under SSP1-RCP1.9 and SSP1-RCP2.6. The two SSP1 scenarios
are under the green growth paradigm. The moderate population growth and fast technological development ease the cropland
demand. Under SSP2-RCP4.5 (Fig. 3c), owing to the implementation of afforestation policy and the improvement of
agricultural production, cropland increment is relatively modest except in some regions such as Africa, South America, and
Southeastern Asia. The most distinctive features of cropland change under SSP3-RCP7.0 (Fig. 3d) is the large expansion in
Western Africa due to the higher food demand, as well as the large reduction in China due to the weak regional mitigation
measures. Scenario SSP4-RCP3.4 (Fig. 3e) is demonstrated to have the largest global cropland expansion, except in some
countries such as Canada, Brazil, and Russia. The rapid population growth and the high mitigation goal improve the demand
for food and biomass energy, resulting in a substantial increase in cropland area. Compared with SSP4-RCP3.4, SSP4-
RCP6.0 includes a more moderate mitigation policy, cropland increments are thus less around the world, especially in Asia,
Central America, and Eastern Europe (Fig. 3f). Both SSP5-RCP3.4OS (Fig. 3g) and SSP5-RCP8.5 (Fig. 3h) follow an SSP5
baseline and exhibit obvious cropland area rise in regions such as Southern America and Africa, but the former has a larger
area increment and also shows a huge increase in places such as the Great Plains of America, Russia, Middle East, and
Southeastern Asia. The overall stronger global cropland expansion under SSP5-RCP3.4OS is due to the large-scale
deployment of bioenergy crops to achieve a lower radiative forcing level.





The above overall cropland changes from history to future are interpretable and logical, and match the qualitative descriptions of cropland changes in some existing studies (Stephens et al., 2019; Ellis et al., 2021; Riahi et al., 2017), indicating the rationality and reliability of our downscaled maps on the whole. To further show the performance of downscaling, we demonstrated some detailed cases (Fig. 4). Several maps at representative time points and places were

selected here, including one of the key origins of agriculture in China (Fig. 4a), European cropland distribution during the Industrial Revolution (Fig. 4b), cropland expansion after the independence of America (Fig. 4c), cropland increase in Brazil under the SSP1-RCP2.6 (Fig. 4d), African cropland growth under the SSP2-RCP4.5 (Fig. 4e), and cropland increase in Southeast Asia under the SSP3-RCP7.0 (Fig. 4f). Although these croplands locate in different biophysical and socioeconomic environments, and their types or appearances vary a lot, they are well characterized in our downscaling maps.

Visually, our downscaling maps match the spatial patterns in HYDE 3.2/LUH2 data and reflect more details of cropland distribution. The spatial heterogeneity and field morphology are clearly characterized in our 1 km maps, whereas for the 10 km or 0.25° maps, it is hard to maintain some small cropland/non-cropland patches. Furthermore, we took the regions shown in Fig. 4c and Fig. 4f as examples to present and compare the cropland distribution of the same places for different years (Fig. 5) or scenarios (Fig. 6). Fig. 5 represents a typical cropland development process in America. The cropland changes from

past to future in the region are coherently and continuously characterized in our downscaled maps. Both cropland increase and decrease can be accurately tracked, and different change amounts and patterns in different locations are well captured. Fig. 6 demonstrates the diverse future cropland development pathways of the region located in Southeast Asia. In our maps, the differences are clearer and vary geographically. Although both regions have varied topography, complex land cover, and fragmented cropland patches, the downscaling results correspond well with the HYDE 3.2/LUH2 and demonstrate fine

details in all these different years or scenarios. The above detailed demonstrations and comparisons also prove the importance of developing higher resolution cropland datasets.

Further quantitative comparisons of spatial distribution between our downscaled cropland data and HYDE 3.2/LUH2 data are presented in Fig. 7. We aggregated the downscaled maps to align the resolution and calculated the correlation coefficient (r) and the Root Mean Squared Error (RMSE) between the corresponding pixels. In general, the datasets exhibited obvious

spatial similarity according to the two indicators. For the historical period, the correlation coefficients are usually lower in previous years especially for pre-1500 CE because of the discrepancy accumulation over time. However, the RMSEs also decline with downscaling time step increases, which is mainly attributed to the reduction of cropland proportion in pixels. For the future period, downscaling performances under different scenarios are distinctive. The most ambitious cropland expansion scenario, SSP4-RCP3.4, displays the minimum consistency. The above inconsistencies are partly influenced by

the base map (Fig. 8), the initial differences transmit into the downscaled maps. The relatively weaker correlations or larger RMSEs in some years or scenarios do not mean our data are incorrect, because our cropland map and HYDE 3.2/LUH2 use different input data and downscaling strategies. But it can indicate the relatively high uncertainty of results, thus, more future efforts are needed for these time periods or scenarios.





## 3.2 Estimated cropland area

According to our downscaled map, on the whole, global cropland area shows an upward trend from 10000 BCE to 2100 CE (Fig. 9). It first steadily and constantly increased from the origin of agriculture. In 1500 CE, there was 2.8 million km$^2$ cropland globally. And cropland area continuously grew to 6.2 million km$^2$ until the beginning of large-scale industrialization (~1850 CE). In 100 years after 1850 CE, the cropland area increment surpassed that during the past 11850 years. In recent decades, the growth rate of cropland slowed down. The area increase in the past 20 years (1990 CE-2010 CE)

has been the smallest since the 18th century. In 2010 CE, the global cropland area was 16.4 million km$^2$. As for the future, the projected cropland areas have substantial discrepancies across the eight SSP-RCP scenarios. Six scenarios (SSP1-RCP1.9, SSP2-RCP4.5, SSP3-RCP7.0, SSP4-RCP3.4, SSP4-RCP6.0, SSP5-RCP3.4OS) yield a monotonously increasing trend with different rates, with the rise ranging from 18.6% to 82.4% between 2010 and 2100. The scenario SSP4-RCP3.4 achieves the largest cropland area increment in the 21st century, more than 50% higher than the scenario with the second-largest

increment (SSP5-3.4OS). For scenarios SSP1-RCP2.6 and SSP5-RCP8.5, the turning points are observed in 2090 CE and 2060 CE, respectively, after which cropland area is expected to decline.

Additionally, we identified large disparities in the cropland areas among different geographical regions (Fig. 10). Countries around the world were divided into five continents to demonstrate their distinctive agricultural development paths. In the early period, the cropland area was higher and grew steadily in Asia, Europe, and Africa. Before 1850 CE, the total cropland

area of these three continents accounts for more than 90% of global cropland area. By contrast, the cropland area in Americas and Oceania did not have substantial increment until the 19th century and the 20th century, respectively. However, after over 200 years of rapid development, Americas become the second largest continent in agricultural land area. In the last decades, except for the accelerated cropland area rise in Africa, the area tends to be stable in the Americas and Asia and even decreased in Europe and Oceania. In the future, cropland areas in Europe and Oceania will experience the smallest changes

in most scenarios (below 0.62 million km$^2$ and 0.25 million km$^2$, respectively), whereas cropland in Africa is projected to maintain evident growth under all scenarios (ranging from 43.6% to 166.4%). Americas and Asia demonstrate different development characteristics under different scenarios and are similar to the global trends mentioned above. The historical and future area trajectories across continents well track historical pathways and future mitigation policies in different human societies, and they are connected logically and smoothly.

To prove the rationality of the results, we also quantitatively compared the cropland area of our dataset with several other datasets (Fig. 11). In years when FAO cropland area statistics (FAO, 2021) were available, our cropland area at continental level was generally consistent in overall patterns with the statistics. The absolute amount of our maps is slightly overestimated for all continents except Europe in several years. The two datasets with full coverage of the entire study period, HYDE 3.2 and LUH2, are taken for the global level comparisons. Our global area time-series are highly correlated to them,

whereas the RMSE and the regression line indicate our results are slightly higher. The above differences are related to the base map (Fig. 12). However, although FAO data and HYDE/LUH2 are widely used, they are not completely correct,





especially in some developing countries with weak agricultural statistics systems (FAO, 2010). In contrast, our selected base map was produced by the fusion of multilevel cropland area statistics and multiple existing cropland maps. It is more reliable in some regions. But regions with large area differences are still noteworthy, more area records and surveys are required to
reduce the uncertainty.

**4 Discussion**

Our downscaled dataset provides spatiotemporally continuous and conceptually consistent global cropland distribution information at 1 km resolution, covering the Holocene period until the end of the 21st century. It coincides well with other well-known land use datasets and exhibits superior detail description. Furthermore, the proposed framework is flexible and
efficient, enabling extensions to specific regions or other land use/cover types. However, there are still some limitations and uncertainties in the study, which are expected to be improved by future research.

First, uncertainties in original demand data, HYDE 3.2 and LUH2, propagated into our downscaled maps. In HYDE 3.2, the total cropland amount for years not covered by FAO statistics (pre-1960 CE) was profoundly determined by modeled population and assumed cropland area per capita curve, which was very uncertain (Klein Goldewijk et al., 2017). In LUH2,
the cropland information was derived from IAM simulations. Errors from the simulations attributed to imperfect model structures and assumptions directly affected the cropland area of LUH2 (Riahi et al., 2017). All these inaccurate original demands directly led to harmonized demand errors and poor downscaling performance, especially in some regions where the differences between the original demands and cropland area of the base map were large. Nevertheless, HYDE 3.2 and LUH2 are regarded as authoritative data and widely used as the basis in related fields despite the above limitations, and there is no
more suitable data to cover such a long time period until now. Moreover, our downscaling work did not focus on simulating the amount of cropland area change, but instead spatially disaggregated the given demands to the grid cells. Nevertheless, there is no doubt that more accurate demands help to get better mapping results. If more reliable cropland area data are available in the future, we can update the results based on them.

Second, the initial cropland map caused some limitations. The downscaling results are greatly dependent on the base map.
The global synergy map was produced based on a series of cropland datasets with various resolutions. Some input data with coarser resolution affected its accuracy. As a result, cropland percentage in the initial cropland map tends to be overestimated in some high-value pixels and underestimated in some low-value pixels. We also tried other cropland maps extracted from several well-known fine-resolution land cover data such as Globeland30 (Chen et al., 2014), Climate Change Initiative Land Cover (CCI-LC) map (ESA, 2017), and Finer Resolution Observation and Monitoring of Global Land Cover
(FROM-GLC) (Gong et al., 2013), however, despite the superior performance in some details, the overall results were even worse. Because of the inconsistent definitions or mapping errors, these satellite-based products generally do not match the statistics and have larger discrepancies with the original demands from HYDE 3.2/LUH2. Therefore, more future efforts should be made to improve the accuracy of cropland maps.




Third, difficulties of data acquisition in suitability evaluation hindered the downscaling. We selected some of the most
common, widely used, and freely available variables for estimating cropland suitability. Nevertheless, they do not represent
all potential factors related to cropland change. Here, we quantified the performance of RF-based suitability evaluation under
the variables acquisition limitations. We randomly collected 0.35 million (accounting for 0.2% global land pixels at 1 km
resolution) test samples worldwide, and we calculated the RMSEs between the 1 km global synergy cropland map and the 1
km suitability layers in 2010 CE. In various WWF biome regions (Fig. 13), the RMSEs are slightly different. At the global
scale, the RMSEs are 15.2% (between biophysical suitability layer and global synergy cropland map) and 14.7% (between
biophysical-socioeconomic suitability layer and global synergy cropland map), respectively, indicating the suitability
evaluation results are acceptable despite the data limitations. Undoubtedly, the performance may get better if more variables
are accessible. In addition to the above variable limitations, the use of population data with resolution coarser than 1 km
partly limited the ability to depict spatial details especially in the early stage of agricultural development (10000 BCE-1500
CE). Besides, the biophysical variables were unavailable beyond the recent decades and remained unchanged in the study.
Therefore, to support more precise downscaling, the related driving factor datasets with high resolution and long time-span
are urgently needed.

Except for the above limitations from input data, others are related to the downscaling model. Some model parameter
settings can affect the results, such as moving window size for kernel density calculation. Since the downscaled dataset in
the study was developed to provide a globally consistent and coherent spatial distribution of cropland, the parameter setting
was based on the situations and rules at the global scale. It did not always incorporate all of the best local data and tune the
parameters for local areas especially. Thus, the value set in the study may not be optimal for some regions. As a result, it can
be applied at the global scale, whereas it cannot be used as the basis for some local research. But the flexible framework
allows researchers to replace input data or revise these model parameters to acquire the best results for their specific study
areas. In addition, similar to lots of other prevalent downscaling models, such as CA (Yang et al., 2015), FLUS (Liao et al.,
2020), Demeter (Le Page et al., 2016), our model simulates the cropland within a downscaling region as a whole. If the total
cropland area in a downscaling region decreases (increases) at the next time step, it is impossible that local non-cropland
(cropland) pixels change to cropland (non-cropland) pixels. As a result, some cropland pixels in historical years that are now
non-cropland and some non-cropland pixels in future years that are now cropland cannot be well captured. In the cropland
downscaling, one of the typical impacts was that our maps could not well characterize the process of cropland being
encroached by growing urban. Therefore, for further model improvement, excepting striving to reduce uncertainties of the
most sensitive parameters, more process details and additional constraints should be included.

## 5 Data availability

The 1 km global cropland maps for the representative years and scenarios shown in Fig.2 and Fig. 3 are available at https://doi.org/10.5281/zenodo.5105689 (Cao et al., 2021a). The complete 1 km global cropland dataset from 10000 BCE to 2100 CE can be viewed at https://cbw.users.earthengine.app/view/globalcroplanddataset. The map values indicate the proportion of cropland within 1 km × 1 km grid cell.

## 6 Conclusions

In the study, the first 1 km resolution global cropland proportion dataset from 10000 BCE to 2100 CE was produced through the proposed harmonization and downscaling framework. Based on our maps, cropland mainly originated in several independent regions, and it gradually spread to other places at various speeds. Some critical historical events affected the global and regional cropland change. As for the future, the cropland distribution is quite different in various scenarios. Globally, the cropland area gradually increases over the past years and displays distinct trends under eight future scenarios.

From 0 million $km^2$ in 10000 BCE, it grows to 2.8 million $km^2$ in 1500 CE, 6.2 million $km^2$ in 1850 CE, and 16.4 million $km^2$ in 2010 CE. Between 2010 CE and 2100 CE, the area growth rate ranges from 18.6% to 82.4%. In different regions, different natural and socioeconomic conditions lead to obvious spatial heterogeneity. Overall, the distribution and area of our cropland maps are consistent with the existing well-known datasets, and it can better characterize the spatial details compared with these datasets. Some small patches and field morphology are more clearly demonstrated. Limitations of the

downscaling originate from the input data and model design, which should be the focus of future research. This high-resolution and long time-span global cropland dataset can support large-scale earth system simulation or detailed agricultural analysis. The harmonization and downscaling framework can be applied in specific regional/local studies or other land use/cover types through the flexible structure and parameters.

## Author contribution

BC and LY conceived the study. All authors contributed to the discussions, writing, and analysis of the manuscript.

## Competing interests

The authors declare that they have no conflict of interest.

## Acknowledgements

This research has been supported by the National Key R&D Program of China (grant nos. 2017YFA0604401 and
2019YFA0606601) and the National Key Scientific and Technological Infrastructure project "Earth System Science Numerical Simulator Facility" (EarthLab).



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



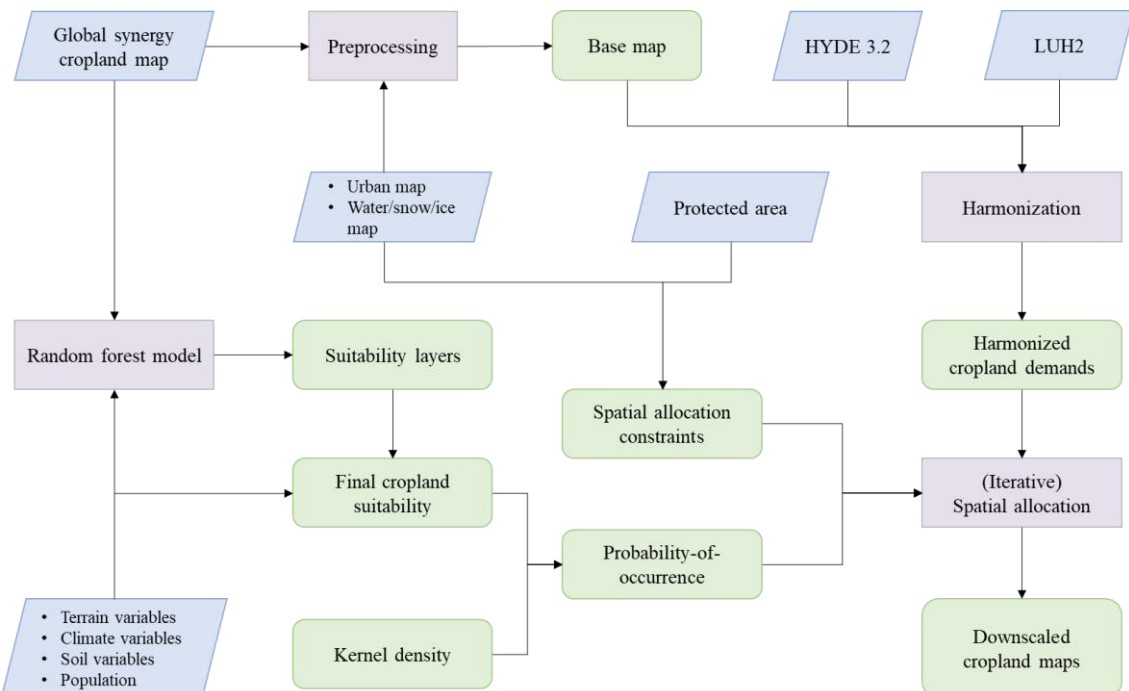

**Figure 1: The framework of producing the 1 km global cropland dataset for 10000 BCE-2100 CE.**



**Table 1: Variables for cropland suitability evaluation.**

| Category | Variables | Year | Resolution | Source |
|---|---|---|---|---|
| **Biophysical variables** | | | | |
| Terrain | Elevation, slope | 2010 CE | 7.5" | Global Multi-resolution Terrain Elevation Data[a] (Danielson and Gesch, 2011) |
| Climate | Mean annual temperature, mean annual precipitation | 1970 CE-2000 CE | 30" | WorldClim version 2.1[b] (Fick and Hijmans, 2017) |
| Soil | Average of soil water, soil PH, and clay content at different depths | 1950 CE-2018 CE | 250 m | OpenLandMap Soil water content at 33kPa[c] (field capacity) (Tomislav and Surya, 2019), OpenLandMap Soil pH in H2O[d] (Tomislav, 2018a), OpenLandMap Clay content[e] (Tomislav, 2018b) |
| **Socioeconomic variables** | | | | |
| Population | population | 10000 CE-2015 CE | 5' | History Database of the Global Environment (HYDE) 3.2[f] (Klein Goldewijk et al., 2017) |

The URLs of these data sources are as follows (last access: 15 July 2021): [a] https://developers.google.com/earth-engine/datasets/catalog/USGS_GMTED2010?hl=en, [b] https://worldclim.org/data/worldclim21.html, [c] https://developers.google.com/earth-engine/datasets/catalog/OpenLandMap_SOL_SOL_WATERCONTENT-33KPA_USDA-4B1C_M_v01?hl=en, [d] https://developers.google.com/earth-engine/datasets/catalog/OpenLandMap_SOL_SOL_PH-H2O_USDA-4C1A2A_M_v02?hl=en, [e] https://developers.google.com/earth-engine/datasets/catalog/OpenLandMap_SOL_SOL_CLAY-WFRACTION_USDA-3A1A1A_M_v02?hl=en, [f] https://doi.org/10.17026/dans-25g-gez3




Figure 2: Downscaled historical cropland maps in (a) 3000 BCE, (b) 1 CE, (c) 1000 CE, (d) 1500 CE, (e) 1700 CE, (f) 1800 CE, (g) 1900 CE, (h) 2000 CE.




**Figure 3: Downscaled future cropland maps in 2100 CE under (a) SSP1-RCP1.9, (b) SSP1-RCP2.6, (c) SSP2-RCP4.5, (d) SSP3-RCP7.0, (e) SSP4-RCP3.4, (f) SSP4-RCP6.0, (g) SSP5-RCP3.4OS, (h) SSP5-RCP8.5.**







Figure 4: Visual comparison between our downscaled maps and HYDE 3.2/LUH2 for six different areas: (a)-(f).



Open Access | Earth System
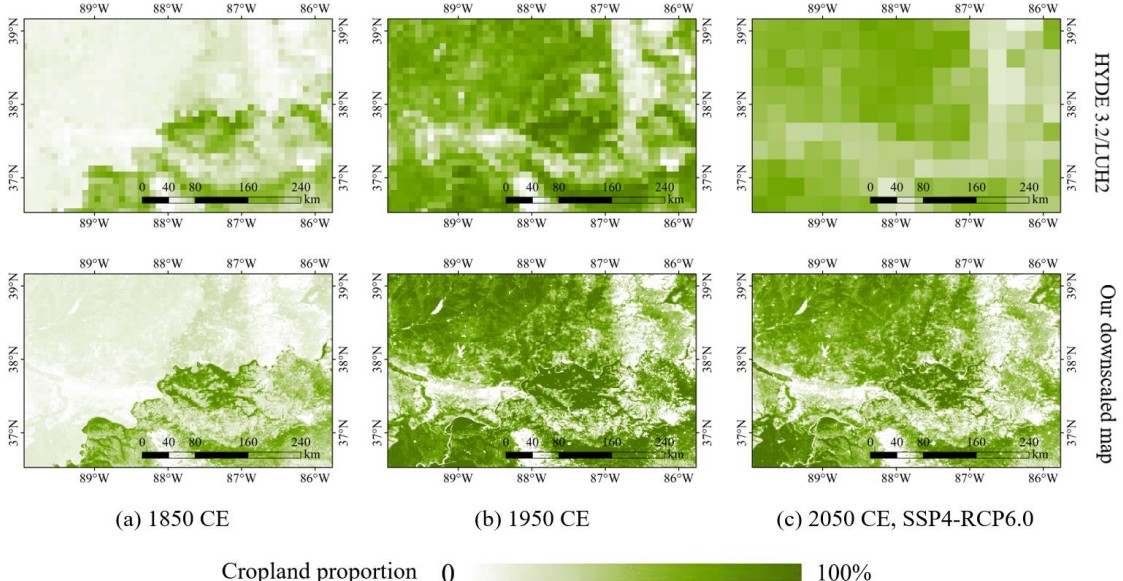

(a) 1850 CE          (b) 1950 CE          (c) 2050 CE, SSP4-RCP6.0

Cropland proportion    0 ▬▬▬▬▬▬ 100%

**Figure 5: Cropland distribution of the region shown in Fig. 4c for different years: (a)-(c).**

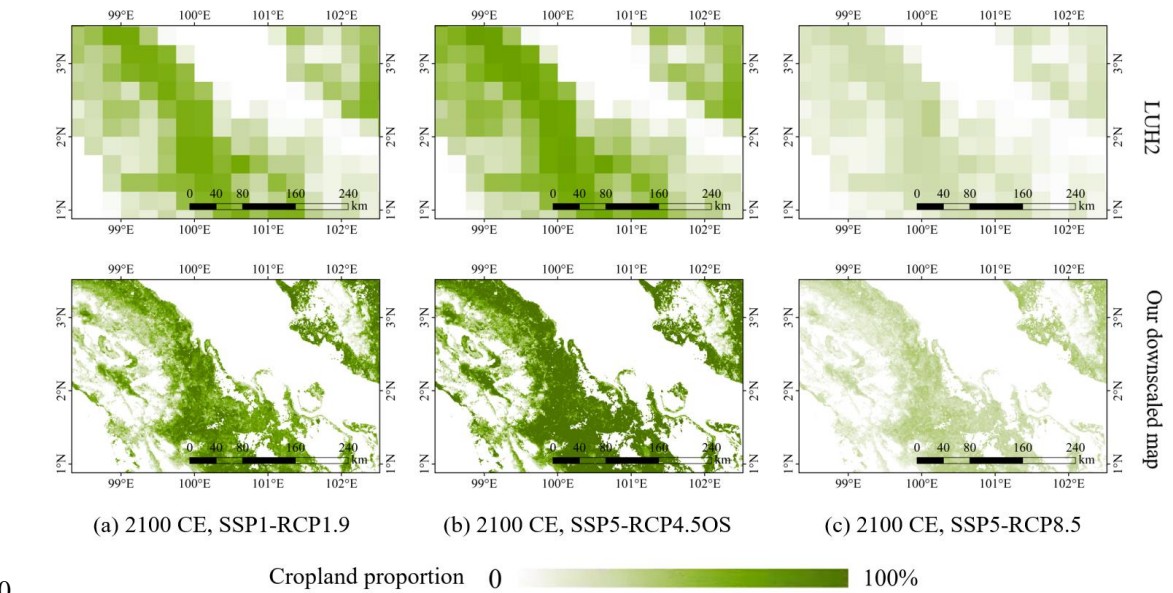

(a) 2100 CE, SSP1-RCP1.9   (b) 2100 CE, SSP5-RCP4.5OS   (c) 2100 CE, SSP5-RCP8.5

Cropland proportion   0 ▬▬▬▬▬▬▬▬▬▬ 100%


**Figure 6: Cropland distribution of the region shown in Fig. 4f for different scenarios: (a)-(c).**

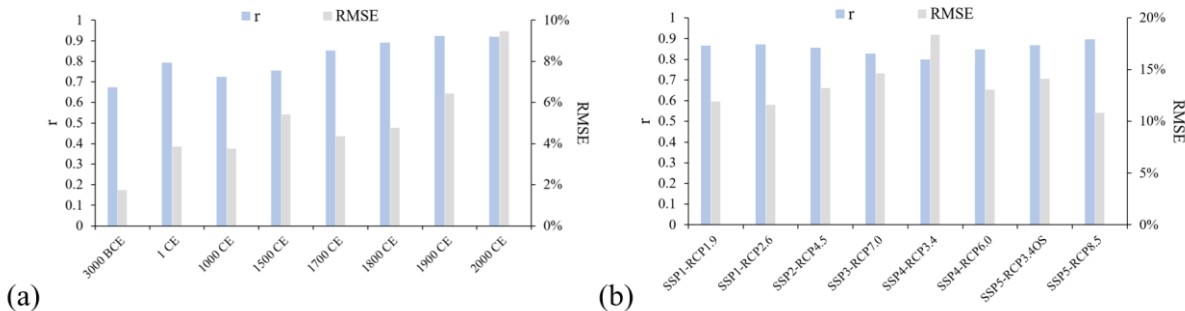

(a)  (b)

**Figure 7: Comparison of cropland proportion in the corresponding pixels between our downscaled map and (a) HYDE 3.2 in the**
**selected eight years, (b) LUH2 in 2100 CE under eight future scenarios. Our downscaled maps were aggregated into 10 km and**
**0.25° resolution for the two comparisons, respectively.**



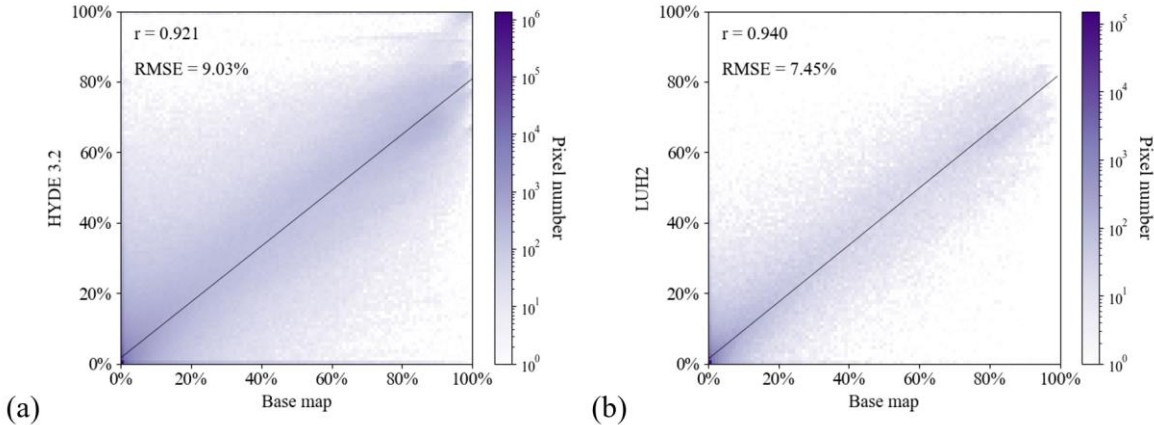

**Figure 8: Comparison of cropland proportion in the corresponding pixels between the base map and (a) HYDE 3.2 in 2010 CE, (b)**
**LUH2 in 2010 CE. Our downscaled maps were aggregated into 10 km and 0.25° resolution for the two comparisons, respectively. The cell value represents the pixel numbers in the corresponding cropland proportion range. The black lines are the linear regression lines.**





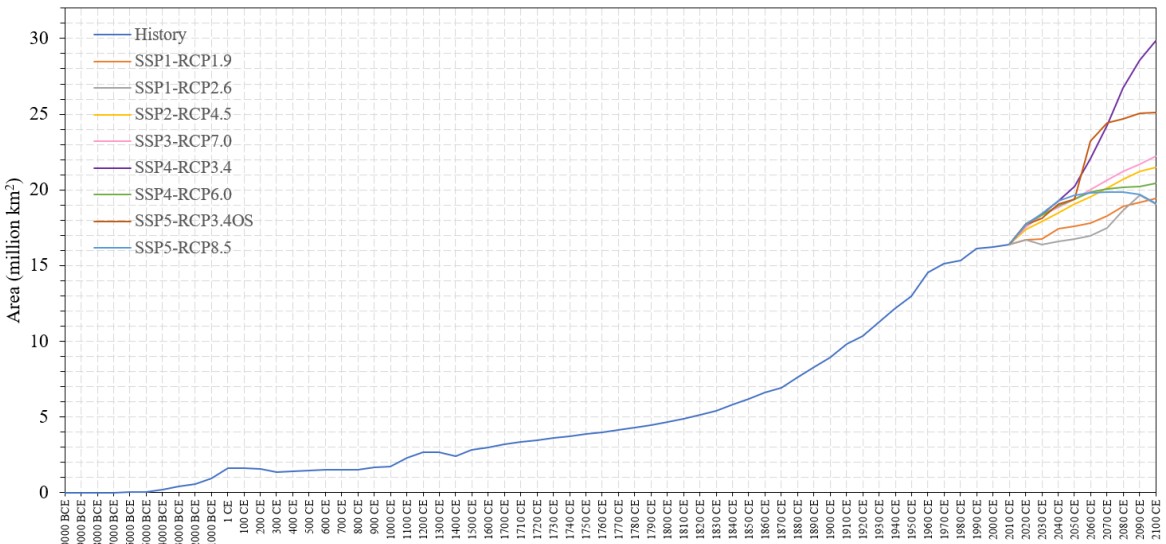

**Figure 9: Global cropland area from 10000 BCE to 2100 CE.**





**Figure 10: Cropland area of different continents for (a) 10000 BCE-2010 CE and (b) 2010 CE-2100 CE under eight future scenarios. The division of continents is based on countries and identical with FAO statistics.**






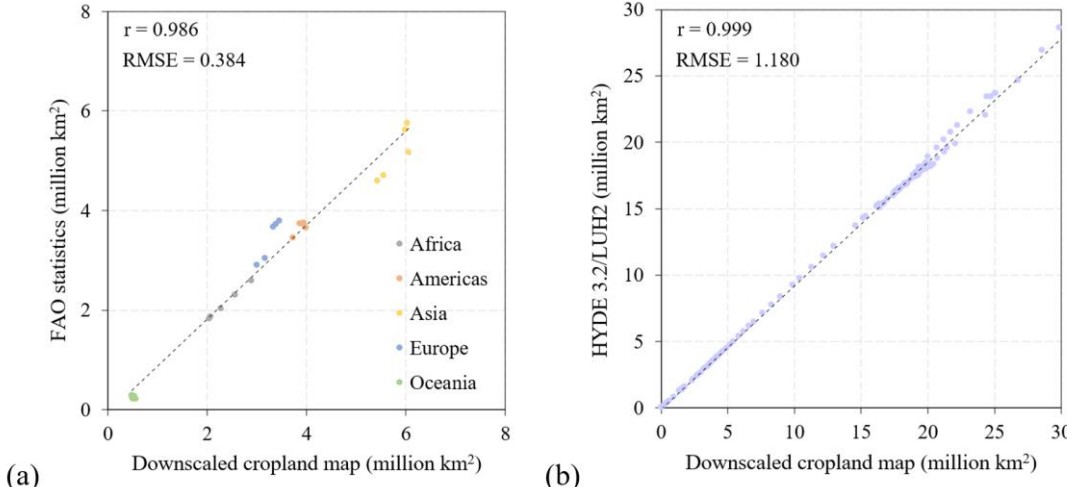

(a)   (b)

**Figure 11: Comparison of cropland area between our downscaled map and (a) FAO statistics at the continental level for 1970 CE-2010 CE, (b) HYDE 3.2/LUH2 at the global level for 10000 BCE-2100 CE under one historical and eight future scenarios. The black lines are the linear regression lines.**




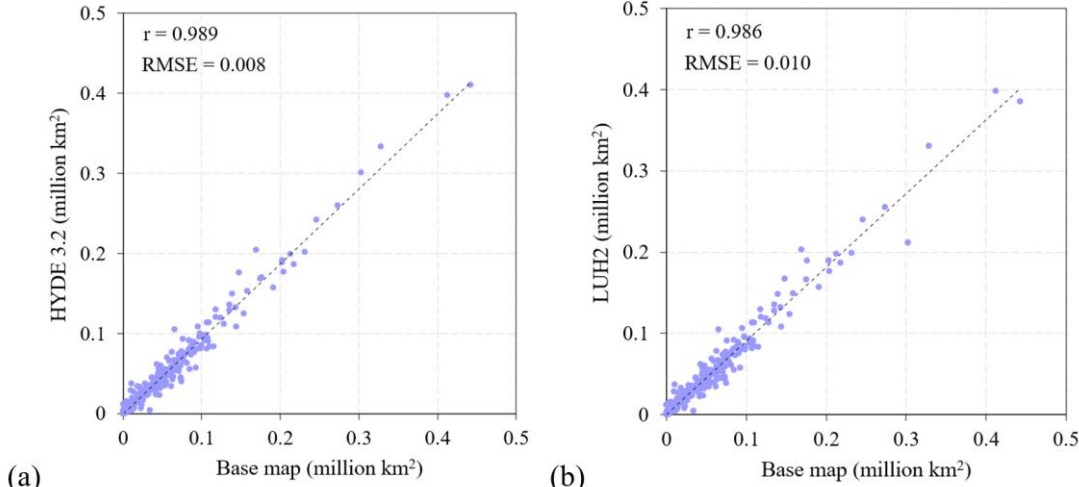

(a)  (b)

**Figure 12: Comparison of cropland area at the downscaling region-level between the base map and (a) HYDE 3.2, (b) LUH2. The black lines are the linear regression lines.**



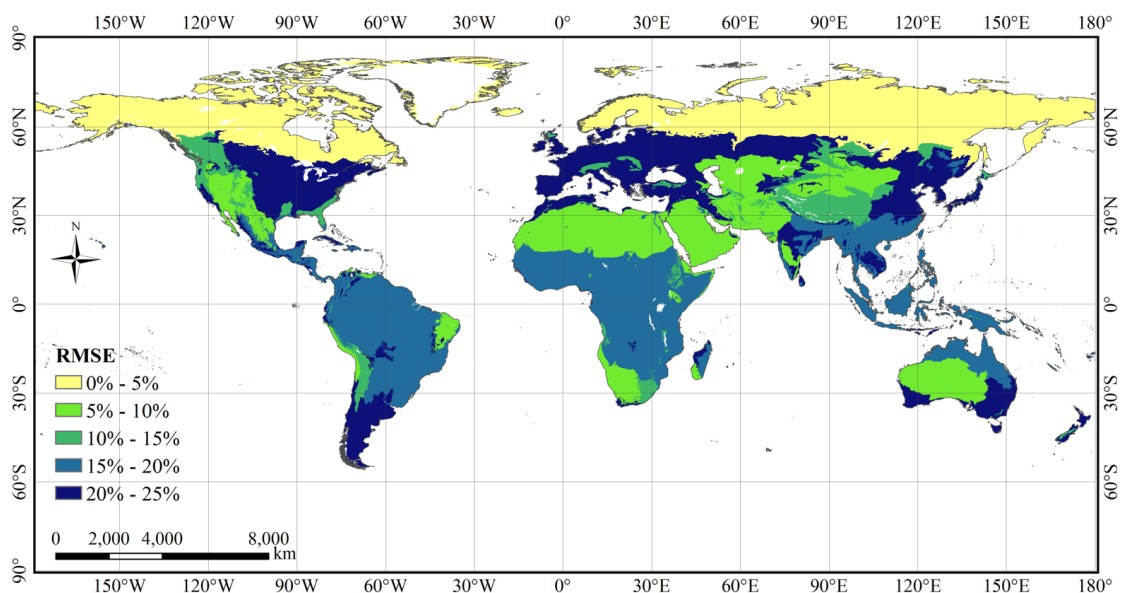


**Figure 13: Random forest-based suitability evaluation performance in different WWF biome regions.**