# Peer review of "A 1 km global cropland dataset from 10000 BCE to 2100 CE"

_Earth System Science Data, 2021_

## Author Comment (AC1)

**Response to comments**

**Title**: A 1 km global cropland dataset from 10000 BCE to 2100 CE

**MS No.**: ESSD-2021-219

**Referee #1**

**General comments**

**Comment 1:**

The contribution of the research is having produced the first 1km global cropland dataset with long time span by employing the newly developed spatially explicit allocation algorithms. The theoretical framework of the method is convincing, and the results are well presented. This spatiotemporally continuous dataset will provide a new opportunity for better understanding the past global changes caused by ALCC.

**Response 1:**

Thank you very much for the comments and suggestions. Please see the detailed point-by-point responses below.

**Specific comments**

**Comment 1:**

Although the new global cropland dataset has been created to such a high resolution (especially for historical periods), it is necessary to recognize the uncertainty of this dataset that was unmentioned in this manuscript. It is suggested to make a brief discussion further to address the issue that both the cropland area per capita estimated by HYDE and the historical population datasets adopted by this study are with huge uncertainty.

Firstly, the reviewer acknowledged that there is no more reliable cropland area data at the global scale than HYDE up to now, but it should be noticed that the historical cropland area data in HYDE have great uncertainty that has already been proved in many countries. One of the main reasons is due to the unreliable estimation of per capita cropland area. The continuous improvement of cropland area data in versions of HYDE mainly relies on the modification of the historical cropland area per capita curve according to the quantitative regional reconstruction results or other related indexes. Although the authors gave a clear definition of cropland (line 118), which was also the same as the definition adopted by HYDE during its historical cropland area estimation. The definition without considering the unignorable amount of fallow land or crop rotation in history would cause obvious underestimation about the cropland areas in some countries. Especially in countries that are far less

intensively cultivated (completely different from the traditional agricultural period in China), like some countries in Europe. Some studies have also pointed out that the area of cultivated land in Europe in HYDE is obviously underestimated. Thus, this would cause a smaller extent of historical cropland distribution and lower fractions in the gridded allocation datasets.

Second, the historical population dataset of HYDE is actually derived from the national or subnational statistical/estimated population amount by downscaling method, which is basically the same as the allocation algorithm of historical cropland. The huge uncertainty also existed in this dataset caused by its allocation principles. Since both studies have adopted the population factor in the allocation algorithm of historical cropland, the difference of gridded datasets between HYDE and this study is namely caused by the different usage of physiogeographic factors and their resolutions during the allocation. In future related researches, please cautiously use this unevaluated historical population dataset.

**Response 1:**

Thank you for your suggestions. We supplemented more uncertainty analysis about the cropland area data in **Section 4, Paragraph 2**: "The cropland area estimation was very sensitive to them, especially the per capita curve shape. The curve construction cannot capture all specific contexts and special events in regional development. Based on credibility assessment using historical facts, regional reconstruction results, and expertise, research has pointed out the cropland area estimation errors in some regions such as Northeast China, North China, and some European countries (Fang et al., 2020)." and uncertainty analysis about population data in **Section 4, Paragraph 4**: "Except for the quantity limitations of variables, the quality limitations of these variable data also impact our results. For instance, the imperfect amount estimation and spatial allocation caused uncertainties of population data (Klein Goldewijk et al., 2017). However, there is no better available substitute. Other variables also have their own uncertainties."

**Reference**

*Fang, X., Zhao, W., Zhang, C., Zhang, D., Wei, X., Qiu, W., and Ye, Y.: Methodology for credibility assessment of historical global LUCC datasets, Sci. China Earth Sci., 63, 1013–1025, https://doi.org/10.1007/s11430-019-9555-3, 2020.*

*Klein Goldewijk, K., Beusen, A., Doelman, J., and Stehfest, E.: Anthropogenic land use estimates for the Holocene - HYDE 3.2, Earth Syst. Sci. Data, 9, 927–953, https://doi.org/10.5194/essd-9-927-2017, 2017.*

**Comment 2:**

Additionally, the cropland results are displayed at a global scale, so the details cannot be seen clearly, and no administrative boundary was added on the regional maps (line 635-640, Fig4-6, it would be better to add some necessary labels and administrative boundaries on the map; it seems that the linear unit scale should not be added under the geographical coordinate system?).

**Response 2:**

Thank you for your comments. We removed the linear unit scale and enlarged the font size of the latitude and longitude labels in **Fig. 4-6**, and we marked locations and coordinates of image center points in the figure titles:

[Figure]

Cropland proportion    0 ▬▬▬▬▬ 100%

**Figure 4: Visual comparison between our downscaled maps and HYDE 3.2/LUH2 for six different areas: (a)-(f). The locations of image center points are as follows: (a) Zhejiang, China (28.338°N, 119.321°E), (b) Adriatic Sea (43.272°N, 14.493°E), (c) Kentucky, America (37.846°N, 87.861°W), (d) Mato Grosso do Sul, Brazil (19.691°S, 54.599°W), (e) Koulikoro Region, Mali (12.105°N, 8.099°W), (f) Riau, Indonesia (2.199°N, 100.425°E).**

[Figure]

Cropland proportion  0 ▭ 100%

**Figure 5: Cropland distribution of the region shown in Fig. 4c for different years: (a)-(c). The location of image center points is Kentucky, America (37.846°N, 87.861°W).**

[Figure]

Cropland proportion  0 ▭ 100%

**Figure 6: Cropland distribution of the region shown in Fig. 4f for different scenarios: (a)-(c). The location of image center points is Riau, Indonesia (2.199°N, 100.425°E).**

---

## Author Comment (AC2)

**Response to comments**

**Title**: A 1 km global cropland dataset from 10000 BCE to 2100 CE

**MS No.**: ESSD-2021-219

**Referee #2**

**General comments**

**Comment 1:**

This study targeted the construction of global cropland dataset starting from 10000 BCE and extending to the future in 2100 CE. Great effort was made to integrate, harmonize, and downscale multi-datasets to produce the final 1 km global dataset. The new dataset is expected to be very useful for a broad spectrum of studies and applications since it considers mapping the historical, as well as future, cropland distribution at a relatively optimal spatial resolution to such a large geographical scale and long period. However, going through the manuscript could raise several questions to the readers, which need to be considered by the authors. Some of those questions are listed as follow.

**Response 1:**

Thank you very much for the comments and suggestions. Please see the detailed point-by-point responses below.

**Specific comments**

**Comment 1:**

Understandably, the date of 10000 BCE was the staring of farming, but integrated datasets (Table 1) were back only to 1950 CE, regardless of the population dataset sourced from HYDE? Hence, defining the suitability for cropland was totally dependent on the population data during the period before 1950, which makes the quality of the cropland mapping during the period totally dependent on the quality of the single layer of the population. Hence, the reader could question the added value and uncertainty when starting the production of the maps from 10000 BCE, with the lack of data covering this long period?

**Response 1:**

Thank you for your comments. During 10000 BCE-1950 CE, the cropland suitability was not only dependent on the population data. As described in Section 2.2, except population, we also used the other variables in suitability evaluation for 10000 BCE-1950 CE. Nevertheless, since these variables are unavailable before 1950 CE, they remained unchanged for these years. It is true that using the unchanged variable data brings some uncertainties. However, it is the commonly used strategy under data

acquisition limitations in land use/cover simulations and there is no better one. Besides, despite the uncertainties, there is a great demand for cropland reconstruction since the start of farming. The complete cropland distribution information throughout the whole process of agriculture development is important and benefits a lot for the overall understanding of agricultural activities. In the study, we also evaluated the uncertainty for the years beyond the recent decades. As shown in Fig. 7, the results are acceptable. As for the added value, compared with HYDE 3.2/LUH2, we used higher resolution variable data and improved methods. According to Fig. 4-6, our maps can better capture the cropland distribution details and spatial heterogeneity, which are very valuable and can serve as a more comprehensive alternative for related applications.

We added the statement in **Section 4, Paragraph 4**: "However, using the unchanged variables is the commonly used strategy under data acquisition limitations in land use/cover simulations and there is no better one. Despite the uncertainties, there is a great demand for complete cropland distribution information throughout the whole process of agricultural development, which is important for the overall understanding of agricultural activities. Compared with HYDE 3.2/LUH2, we used higher resolution data and improved methods. Even if for the years beyond the recent decades, our maps can still better capture the cropland distribution details and spatial heterogeneity (Fig. 4-6). In the study, we also evaluated the uncertainty for these years and the results are acceptable (Fig. 7)."

**Comment 2:**

Suitability map played a crucial role in mapping production. The authors clearly stated that the influence of the variables defining the land suitability for agriculture was not equal throughout the whole period. For instance, population was the key variable defining the land suitability in earlier dates due to the traditional farming practices and weak global links (as mentioned in line 169). However, another assumption in line 175 was made by authors that I believe it needs further explanation and justification. The assumption is that "the impact of population on cropland distribution is negligible for most regions in the future years (2010 CE-2100 CE)", why?

**Response 2:**

Thank you for your comments. We agree that there is still a relationship between total future global cropland area and total future global population. However, the relationship between cropland area and population will become much weaker when it comes to finer grid scales (Meyer and Turner II, 1992). Future technology development will weaken the population demand in crop farming, making crop farming no longer heavily rely on population. Moreover, future trade activities will become more frequent. The demanded crops of many regions will be not produced locally but from other regions. Besides, the population continues to migrate and gather to urban, resulting in stronger population intensification. In some regions especially metropolises, although population increases a lot, the cropland area increases little or even decreases. Thus, at grid-scale, the

60    indication of future population change on future cropland change will become weak for lots of regions worldwide. Additionally, there is no suitable future population dataset. Therefore, we exclude the population variable in future cropland simulations.

We added the further explanation in **Section 2.2**: "In the future (2010 CE-2100 CE), technology development will make crop farming no longer heavily rely on population, and trade boom will enable many regions to meet the crop demands of the local population through purchasing from other regions. Besides, future population intensification tends to be stronger. Thus, at the

65    fine grid-scale, the relationship between cropland area and population will become much weaker for lots of regions worldwide in the future. Additionally, there is no suitable future population dataset."

**Reference**

*Meyer, W. B. and Turner II, B. L.: Human population growth and global land-use/cover change, Annu. Rev. Ecol. Syst., 23, 39–61, https://doi.org/10.1146/annurev.es.23.110192.000351, 1992.*

70

**Comment 3:**

The comparison between the constructed cropland dataset and HYDE 3.2 dataset in Figure 7A showed a relatively close r value for recent years (1700-2000) but a high increase of RMSE value. A reader could expect that the error in quantifying

75    cropland area would decrease when getting closer to the current time. But this is not the case in the constructed dataset, any justification from the authors about that?

**Response 3:**

Thank you for your comments. As interpreted in Section 3.1, "the RMSEs also decline with downscaling time step increases (i.e., from present to past), which is mainly attributed to the reduction of cropland proportion in pixels".

80    The RMSE is calculated based on the cropland proportion differences of the corresponding pixels between our dataset and HYDE 3.2. It is directly affected by the absolute values of cropland proportion. The cropland proportions are generally lower for the early stage, so, the RMSEs are also smaller. It should be noted that the RMSE is scale-dependent since it represents absolute differences rather than relative differences. Therefore, direct comparisons of RMSEs between different years/scenarios are actually invalid due to the different scales. Besides, the larger RMSE values do not represent larger errors,

85    but only indicate greater differences between our data and HYDE 3.2.

We supplemented the statement in **Section 3.1**: "It should be noted that the RMSE is scale-dependent since it represents absolute differences rather than relative differences. Therefore, direct comparisons of RMSEs between different years/scenarios are actually invalid due to the different scales."